# Prescribe-then-Select: Adaptive Policy Selection for Contextual Stochastic Optimization

**Caio de Próspero Iglesias** *caiopigl@mit.edu*
*Sloan School of Management*
*Massachusetts Institute of Technology*

**Kimberly Villalobos Carballo** *kimberly.v@nyu.edu*
*Tandon School of Engineering*
*New York University*

**Dimitris Bertsimas** *dbertsim@mit.edu*
*Sloan School of Management*
*Massachusetts Institute of Technology*

**Reviewed on OpenReview:** *https://openreview.net/forum?id=lFEsAF2I7C*

## Abstract

We address the problem of policy selection in contextual stochastic optimization (CSO), where covariates are available as contextual information and decisions must satisfy hard feasibility constraints. In many CSO settings, multiple candidate policies—arising from different modeling paradigms—exhibit heterogeneous performance across the covariate space, with no single policy uniformly dominating. We propose Prescribe-then-Select (PS), a modular framework that first constructs a library of feasible candidate policies and then learns a meta-policy to select the best policy for the observed covariates. We implement the meta-policy using ensembles of Optimal Policy Trees trained via cross-validation on the training set, making policy choice entirely data-driven. Across two benchmark CSO problems—single-stage newsvendor and two-stage shipment planning—PS consistently outperforms the best single policy in heterogeneous regimes of the covariate space and converges to the dominant policy when such heterogeneity is absent. All the code to reproduce the results can be found at https://github.com/iglesiascaio/Prescribe-then-Select.

## 1 Introduction

Optimization under uncertainty arises in numerous applications, including supply chain management, energy markets, and financial planning, and remains an important research area in the optimization community (Birge & Louveaux, 2011; Shapiro et al., 2021; Conejo et al., 2010; Snyder, 2006; Snyder & Shen, 2019). In these problems, decisions are made under uncertainty about future outcomes, with the objective of minimizing total cost. A wide variety of data-driven methods have been developed to address such problems, including sample average approximation (Shapiro, 2003; Shapiro & Nemirovski, 2005; Kleywegt et al., 2002), stochastic approximation algorithms (Robbins & Monro, 1951; Nemirovski et al., 2009), robust optimization techniques (Bertsimas et al., 2018b; Ben-Tal et al., 2009; Bertsimas et al., 2018a), and distributionally robust optimization formulations (Delage & Ye, 2010; Calafiore & Ghaoui, 2006). These approaches differ in how they model and mitigate uncertainty: some aim to minimize expected costs (e.g., sample average or stochastic approximation methods), while others focus on guarding against worst-case outcomes (as in robust optimization) or balance the two by minimizing worst-case expected cost under a range of plausible distributions (as in distributionally robust optimization).

In many real-world settings, however, the decision-maker also observes auxiliary data—known as covariates, context, or side information—prior to making a decision. For example, in transportation, real-time weather

and traffic conditions can help reduce uncertainty in travel times; in retail, promotions and seasonal signals can inform demand; and in healthcare, patients' comorbidities and clinical histories provide valuable information about health outcomes. When such covariates are available, they can help infer relevant characteristics of the underlying uncertainty, narrowing the set of plausible scenarios and enabling more informed decisions. This observation has led to the development of contextual stochastic optimization (Sadana et al., 2025), where the goal is to learn a policy that maps covariates to decisions.

A diverse set of methodologies has emerged for solving contextual stochastic optimization problems. One approach is to fit a parametric decision rule—such as a linear function (Ban & Rudin, 2019; Bertsimas & Kallus, 2020), kernel-based model (Bazier-Matte & Delage, 2020; Ban & Rudin, 2019; Bertsimas & Koduri, 2022; Notz & Pibernik, 2022; Bertsimas & Carballo, 2023), or neural network (Oroojlooyjadid et al., 2020; Huber et al., 2019; Zhang & Gao, 2017)—by minimizing empirical cost (potentially regularized) over historical data. These methods are computationally efficient at deployment since no optimization is needed once the policy is trained. However, their performance is often sensitive to the choice of function class and may suffer if the true policy is poorly approximated within the chosen function space. Importantly, these approaches are not suitable for constrained CSO problems, where feasibility constraints—potentially even integrality constraints—must be satisfied for every decision.

An alternative family of methods, commonly referred to as predict-then-optimize, follows a two-stage approach: first, the conditional distribution of the uncertain parameters is estimated given the observed covariates; then, this estimate is used to approximate the expected cost and solve the corresponding optimization problem (Bertsimas & Kallus, 2020; Ban & Rudin, 2019; Kannan et al., 2024; Deng & Sen, 2022). Such two-stage approaches offer modularity and interpretability; yet, they risk suboptimal decisions when accurate predictions do not translate into good prescriptions—particularly when the loss function used in training is misaligned with the downstream cost. To mitigate this, recent work has focused on end-to-end training schemes that integrate optimization objectives with predictive modeling (Bengio, 1997; Donti et al., 2017; Kallus & Mao, 2023; Qi et al., 2021; Elmachtoub & Grigas, 2022). These methods take into account the optimization task loss during predictive model training. However, they introduce new challenges, such as increased computational burden during training. In addition, they have been shown to often achieve substantially slower regret convergence rates compared to the former non-integrated approach (Hu et al., 2022).

Each of these paradigms presents trade-offs between statistical guarantees, computational complexity, and decision quality. In particular, when multiple methods or policies are available—perhaps developed using different paradigms, modeling choices, or feature sets—an important question arises: how should one select among a set of candidate policies? In this paper, we address this question through the following contributions:

1. We formalize the problem of policy selection in contextual stochastic optimization, where covariates are available as contextual information, and decisions must satisfy hard feasibility constraints.

2. We introduce *Prescribe-then-Select*, a modular framework that first generates a diverse library of feasible candidate policies and then uses supervised learning to train a meta-policy that maps each covariate to the best candidate policy.

3. We conduct extensive computational experiments on two benchmark CSO problems in operations management: newsvendor and shipment planning. Results show that the proposed meta-policy improves over the best single policy in settings where different policies perform best across different regions of the covariate space, and matches the best single policy otherwise.

## 2 Related Work

Our work relates to several strands of literature on selecting among a set of candidate decision policies in the presence or absence of contextual information. While there is extensive research on policy ensembling, mixture-of-experts architectures, and divide-and-conquer strategies in reinforcement learning (RL), the problem has received far less attention in the setting of contextual optimization under uncertainty within operations research, where feasibility constraints and explicit cost function minimization are critical.

One line of work in reinforcement learning examines policy ensembling in context-free settings, where multiple policies are combined to improve stability, robustness, or exploration without conditioning on the current context (Wiering & Van Hasselt, 2008; Duell & Udluft, 2013). These methods typically aggregate policies by averaging their action outputs or using majority voting, aiming to reduce variance, mitigate overfitting, and hedge against the weaknesses of individual learners. While effective in improving average performance, these methods do not adapt policy choice to specific states, tasks, or other contextual information, and therefore cannot exploit scenarios where different policies are optimal in different contexts.

To address such heterogeneity, RL has developed context-dependent policy selection mechanisms, most prominently through mixture-of-experts architectures. Here, a meta-policy—often referred to as a gating function—maps the current context to a distribution over expert policies. In early work (Doya et al., 2002; Samejima et al., 2003; Van Seijen et al., 2008), the context was simply the state of the environment, leading individual policies to specialize in different regions of the state space; the meta-policy would then either combine their actions (soft selection) or choose the highest-scoring policy (hard selection). More recently, Gimelfarb et al. (2021) extended this approach to contextual policy transfer, in which the context incorporates higher-level task descriptors in addition to the state, allowing the meta-policy to select from a library of policies that were trained on different tasks but may transfer knowledge to the current task. Related approaches in divide-and-conquer RL partition the state space and assign a policy to different regions (Ghosh et al., 2017; Goyal et al., 2019). While these techniques achieve context-dependent specialization, they are typically studied in online settings and operate in unconstrained action spaces where policies can be aggregated without violating feasibility. Moreover, partitions of the contextual space are often learned to optimize surrogate objectives that are disconnected from the true cost function of interest in the downstream optimization problem.

In contrast, contextual optimization under uncertainty has seen almost no work on combining or selecting from multiple policies. The only example we are aware of is Cui et al. (2025), who recently proposed learning a fixed linear combination of candidate policies to improve decision quality. Their method uses historical data to estimate weights for each policy and forms a weighted sum of their outputs, demonstrating performance gains in the unconstrained newsvendor problem (Khouja, 1999). While promising in unconstrained settings, this approach is not generally applicable when decisions must satisfy hard feasibility constraints, since averaging feasible policies does not necessarily produce another feasible policy. Therefore, there remains a need for methods that leverage contextual information to select the best policy from feasible candidates.

In this paper we study the problem of policy selection in single-stage and two-stage contextual optimization under uncertainty, where decisions do not affect the underlying uncertainty and must satisfy hard feasibility constraints. We propose a general framework which first generates a library of feasible candidate policies, and then learns a supervised selection model to choose exactly one policy for each instance based on its context. By tailoring the policy assignment to each observed covariate, the approach can outperform any single candidate policy on average, especially when different policies are optimal in different regions of the covariate space. This enables flexible reuse of high-performing policies, adaptation to heterogeneous contexts, and robust performance across complex decision environments.

The remainder of the paper is organized as follows. Section 3 introduces the general formulation of contextual optimization under uncertainty, and Section 4 presents the proposed *Prescribe-then-Select* framework. Section 5 reports the results of our computational experiments, and Section 6 offers concluding remarks.

## 3 Problem Setting

We consider the problem of *contextual stochastic optimization* (CSO) (Sadana et al., 2025), in which a random vector of covariates $X \in \mathcal{X} \subseteq \mathbb{R}^{d_x}$—capturing context such as holidays, promotions, or seasonal trends—is observed as side information in a stochastic optimization problem. After observing $X = \boldsymbol{x}$, the decision-maker selects an action $\boldsymbol{z}$ from a feasible set $\mathcal{Z} \subseteq \mathbb{R}^{d_z}$ representing all constraints that the decision $\boldsymbol{z}$ must satisfy. The objective is to minimize the expected cost with respect to some unknown random variable $Y \in \mathcal{Y} \subseteq \mathbb{R}^{d_y}$. We assume that a cost function $c : \mathcal{Z} \times \mathcal{Y} \to \mathbb{R}$ is provided, and that decisions $\boldsymbol{z}$ do not affect

the uncertainty $Y$. Specifically, the problem can be written as

$$v^\star(\boldsymbol{x}) = \min_{\boldsymbol{z} \in \mathcal{Z}} \ \mathbb{E}\big[c(\boldsymbol{z}, Y) \mid X = \boldsymbol{x}\big], \qquad \pi^\star(\boldsymbol{x}) \in \arg\min_{\boldsymbol{z} \in \mathcal{Z}} \ \mathbb{E}\big[c(\boldsymbol{z}, Y) \mid X = \boldsymbol{x}\big], \tag{1}$$

where $v^\star$ is the optimal value function and $\pi^\star$ is the optimal policy function. Given a dataset $\mathcal{D}_N = \{(\boldsymbol{x}_i, \boldsymbol{y}_i)\}_{i=1}^N$ with historical observations, the data-driven approach to solving equation 1 consists of using these observations to approximate the conditional expectation in the objective.

Two classical baselines illustrate the spectrum of data-driven approaches. At one extreme, the *sample-average-approximation* (see, e.g., Kleywegt et al. (2002)) method ignores covariates altogether, replacing the conditional expectation in equation 1 with its unconditional counterpart and approximating it via the empirical average. While simple to implement, sample-average approximation produces the same decision for all contexts, even when covariates could help narrow down the realized value of the uncertainty $Y$. At the other extreme lies the *point-prediction* approach (see, e.g., Bertsimas & Kallus (2020)), which fits a predictive model $\hat{\mu}_N(\boldsymbol{x})$ to estimate $\mathbb{E}[Y \mid X = \boldsymbol{x}]$, and then substitutes this prediction into the objective, converting the problem into $\arg\min_{\boldsymbol{z} \in \mathcal{Z}} c\big(\boldsymbol{z}, \hat{\mu}_N(\boldsymbol{x})\big)$. This method tailors decisions to the observed context and is computationally efficient, but it reduces the entire conditional distribution to a single point estimate, ignoring its variance and how the cost function responds to uncertainty beyond the mean.

An important obstacle in CSO problem is that $X$ and $Y$ can be highly variable across regimes. For instance, in retail sales, routine weekdays, promotional weekends, and holiday peaks may each exhibit very different conditional demand distributions $f_{Y|X}(y \mid x)$. Such heterogeneity in the covariate space may also induce heterogeneity in policy performance, with different policies performing best in different covariate regimes. For instance, low-variance regions often reward the precision of point-prediction, whereas high-variance regions may benefit from approaches that explicitly account for uncertainty in $c(\boldsymbol{z}, Y)$ such as sample average approximation. Likewise, many other existing approaches to the CSO problem may perform well in certain regions of the covariate space but not on average across the entire space. Figure 1 illustrates this phenomenon for the classical newsvendor problem, where different candidate policies dominate in different segments of the covariate space, an empirical result examined more thoroughly in Section 5.4.1. These observations motivate the use of multiple candidate policies, each designed to perform well under a specific regime, together with a selection mechanism that assigns the most cost-effective policy to each context. The remainder of the paper develops and analyzes this approach.

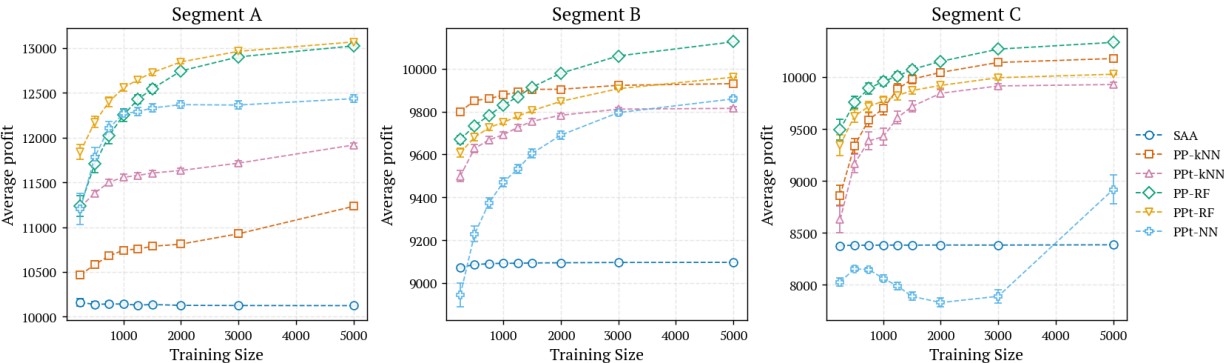

Figure 1: Newsvendor: segment-wise profit vs. training size (average and 95% CIs shown).

## 4 Methodology: Prescribe-then-Select

This section presents our proposed method for addressing covariate heterogeneity in contextual stochastic optimization by combining multiple decision policies within a unified framework. Instead of relying on a single model across the entire covariate space, we construct a library of diverse candidate policies and use a meta-policy to select one among them based on the observed covariates. Each candidate policy maps contextual features to feasible decisions under distinct modeling assumptions and inference mechanisms. They may differ in the type of predictive estimates they produce, their complexity, or their inductive biases.

The meta-policy then leverages this diversity by jointly learning a partition of the covariate space and deciding the most effective policy for each region.

We illustrate the methodology using a few representative examples of individual policies that often exhibit heterogeneous performance across the covariate space. The framework, however, is readily extensible and can incorporate policies developed under other approaches to the CSO problem. We obtain the meta-policy by training an ensemble of policy trees, which are depth-constrained decision trees over the covariate space that aim to minimize the empirical cost of the CSO problem over historical data. We refer to the proposed method as Prescribe-then-Select, reflecting its two-stage structure: first prescribing a set of candidate policies, then selecting the most suitable one for each region of the covariate space. The following subsections describe each component of this method in detail.

## 4.1 Prescribing: Developing a library of candidate policies

The first stage of our framework constructs a library of prescriptive models, each representing a policy that makes feasible decisions given the observed covariates. We here consider three families of prescriptive methods: sample-average-approximation policies, point–prediction policies, and predictive–prescriptive policies. However, we highlight that our framework could include any other feasible policies to the CSO problem.

### 4.1.1 Sample Average Approximation (SAA).

The sample-average-approximation method (Kleywegt et al., 2002) is a classical approach for solving stochastic optimization problems by replacing the true expectation in equation 1 with an empirical average computed from a finite sample. Given $N$ historical scenarios $\{\boldsymbol{y}_i\}_{i=1}^N$, each representing a realization of the uncertain parameters, SAA approximates the expectation as $\mathbb{E}[c(\boldsymbol{z}, Y)] \approx \frac{1}{N} \sum_{i=1}^N c(\boldsymbol{z}, \boldsymbol{y}_i)$ and finds a prescription by solving the following optimization problem:

$$\pi^{\text{SAA}} \in \underset{\boldsymbol{z} \in \mathcal{Z}}{\arg\min} \ \frac{1}{N} \sum_{i=1}^N c(\boldsymbol{z}, \boldsymbol{y}_i). \tag{2}$$

Unlike other CSO policies, SAA does not condition on covariates $\boldsymbol{x}$ and instead treats all observed scenarios equally. As a result, it is well suited to settings where covariate information is absent or irrelevant. Solving equation 2 reduces the problem to a single deterministic optimization problem, making SAA straightforward to implement.

### 4.1.2 Point–Prediction Policies (PPt).

The point–prediction framework first fits a predictive model $\hat{\mu}_N(\boldsymbol{x}) \approx \mathbb{E}[Y \mid X = \boldsymbol{x}]$ and then solves a deterministic optimization problem where uncertainty is collapsed to its predicted mean:

$$\pi^{\text{PPT}}(\boldsymbol{x}) = \underset{\boldsymbol{z} \in \mathcal{Z}}{\arg\min} \ c\big(\boldsymbol{z}, \hat{\mu}_N(\boldsymbol{x})\big). \tag{3}$$

This approach is efficient and interpretable, and often performs well when the cost function is approximately linear and the conditional distribution of $Y$ has low variance. We illustrate this approach with two predictive models:

**(a) PPt-kNN:** Given a data point $\boldsymbol{x} \in \mathcal{X}$, the k-nearest neighbor model (kNN) (Cover & Hart, 1967) identifies the k training samples $\{(\boldsymbol{x}_i, \boldsymbol{y}_i)\}$ closest to $\boldsymbol{x}$ under a fixed distance metric. Denoting by $\mathcal{N}_{\text{k}}(\boldsymbol{x})$ the set of indices $i$ for these neighbors, we use $\hat{\mu}_N(\boldsymbol{x}) = \frac{1}{\text{k}} \sum_{i \in \mathcal{N}_{\text{k}}(\boldsymbol{x})} \boldsymbol{y}_i$ in the case of numerical uncertainty, or majority vote in the case of categorical uncertainty.

**(b) PPt-RF:** Random forests (RF) are ensemble models that aggregate predictions from multiple decision trees, each trained on a different bootstrap sample of the data and typically using randomized feature selection at each split (Breiman, 2001). This procedure, known as bagging (bootstrap aggregation), reduces variance and enhances generalization. Let $T_1, \ldots, T_B$ denote the trees in the forest, and let $\mathcal{L}_b(\boldsymbol{x})$ be the set of training indices falling in the same leaf as $\boldsymbol{x}$ in tree $T_b$. The predictions for a given input $\boldsymbol{x}$ are then given by $\hat{\mu}_N(\boldsymbol{x}) = \frac{1}{B} \sum_{b=1}^B \frac{1}{|\mathcal{L}_b(\boldsymbol{x})|} \sum_{i \in \mathcal{L}_b(\boldsymbol{x})} \boldsymbol{y}_i$.

**(c) PPt-NN:** A feed-forward neural network (NN) (Goodfellow et al., 2016) is defined as a nonlinear map $f_{\boldsymbol{\theta}} : \mathcal{X} \to \mathbb{R}^{d_y}$ that recursively applies affine transforms and non-linear activation functions. Starting from $\boldsymbol{h}^{(0)} = \boldsymbol{x}$, each layer computes $\boldsymbol{h}^{(\ell)} = \phi(\boldsymbol{W}^{(\ell)}\boldsymbol{h}^{(\ell-1)} + \boldsymbol{b}^{(\ell)})$ where $\phi$ is the activation function (e.g., ReLU, tanh) for all $\ell \in [L]$; and finally outputs $f_{\boldsymbol{\theta}}(\boldsymbol{x}) = \sigma(\boldsymbol{h}^L)$ where $\sigma$ is the identity function for regression tasks or the softmax function for classification tasks. Parameters $\boldsymbol{\theta} = \{\boldsymbol{W}^{(\ell)}, \boldsymbol{b}^{(\ell)}\}_{\ell \in [L]}$ are typically fit via empirical risk minimization with stochastic gradient methods. The PPt prescription sets $\hat{\mu}_N(\boldsymbol{x}) = f_{\boldsymbol{\theta}}(\boldsymbol{x})$.

### 4.1.3 Predictive–Prescriptive Policies (PP).

The predictive–prescriptive framework (Bertsimas & Kallus, 2020; Bertsimas & Dunn, 2019) integrates local statistical estimation with scenario-based optimization, and can be interpreted as a form of weighted sample-average-approximation (Sadana et al., 2025). Given a predictive model that assigns weights $w_{N,i}(\boldsymbol{x})$ to each training pair $(\boldsymbol{x}_i, \boldsymbol{y}_i)$, this approach approximates the conditional expectation in equation 1 as

$$\mathbb{E}\big[c(\boldsymbol{z}, Y) \mid X = \boldsymbol{x}\big] \approx \sum_{i=1}^{N} w_{N,i}(\boldsymbol{x})\, c(\boldsymbol{z}, \boldsymbol{y}_i), \quad \text{and solves} \quad \pi^{\mathrm{PP}}(\boldsymbol{x}) \in \arg\min_{\boldsymbol{z} \in \mathcal{Z}} \sum_{i=1}^{N} w_{N,i}(\boldsymbol{x})\, c(\boldsymbol{z}, \boldsymbol{y}_i). \tag{4}$$

These normalized weights ($\sum_{i=1}^{N} w_{N,i}(\boldsymbol{x}) = 1$) capture the local distributional information implied by each model and are used to generate prescriptions tailored to the covariate $\boldsymbol{x}$. Using the PP framework, the kNN and RF predictive models would lead to the following weighting schemes:

**(a) PP-kNN:** In this method, the nearest neighbors of a data point are not used to estimate $\mathbb{E}[Y \mid X = \boldsymbol{x}]$, but rather they are used to estimate a local approximation of the conditional distribution $f_{Y|X}(\cdot \mid \boldsymbol{x})$. Specifically, the weights in the PP approach are given by:

$$w_{N,i}^{\mathrm{kNN}}(\boldsymbol{x}) = \begin{cases} \frac{1}{k}, & \text{if } i \in \mathcal{N}_{\mathrm{k}}(\boldsymbol{x}), \\ 0, & \text{otherwise.} \end{cases}$$

**(b) PP-RF:** For a given data point $\boldsymbol{x} \in \mathcal{X}$, each tree in the forest routes $\boldsymbol{x}$ to a leaf node. The collection of training samples retrieved from the leaf nodes across all trees forms a local, ensemble-based approximation to the conditional distribution $f_{Y|X}(\cdot \mid \boldsymbol{x})$. As with kNN, this distribution can be used to support scenario-based optimization. The PP approach in this case defines the weights as:

$$w_{N,i}^{\mathrm{RF}}(\boldsymbol{x}) = \frac{1}{B} \sum_{b=1}^{B} \frac{\mathbb{1}[i \in \mathcal{L}_b(\boldsymbol{x})]}{|\mathcal{L}_b(\boldsymbol{x})|}.$$

## 4.2 Selecting: Learning the best policy for each context

The second stage of our framework learns a meta-policy that chooses the best candidate policy for the given context. Let the library of candidate policies be denoted by $\Pi_M = \{\pi^m\}_{m=1}^{M}$, each mapping covariates $\boldsymbol{x} \in \mathcal{X}$ to a feasible decision $\boldsymbol{z} \in \mathcal{Z}$. To learn the meta-policy we adopt *Optimal Policy Trees* (OPT) (Amram et al., 2022), which partitions the feature space into axis-aligned regions and assigns a fixed policy index to each region. While extensions that allow for more general splits—such as linear combinations of features—do exist, we do not explore them in this paper.

OPTs solve a treatment assignment problem from observational data. The OPT method learns a decision tree that maps covariates to the treatment that optimizes expected outcomes. Importantly, we do not use the OPT to obtain a feasible policy for the CSO problem, since just like parametric decision rules, the OPT could not be tractably optimized to satisfy hard constraints on the decisions. We instead leverage OPTs to select among a set of feasible candidate policies based on the observed covariates. In other words, we consider each candidate policy as a treatment; and train the OPTs to learn which treatment is best given the contextual information.

Formally, a policy tree is a decision tree $T(x; \Theta)$, parameterized by $\Theta = \{(R_j, \gamma_j)\}_{j=1}^{J}$, where $\{R_j\}_{j=1}^{J}$ is a disjoint partition of $\mathcal{X}$ and $\gamma_j \in [M]$ denotes the policy index assigned to region $R_j$, with $[n] := \{1, \ldots, n\}$. The policy induced by this tree is $\pi^{T(\boldsymbol{x};\Theta)}(\boldsymbol{x})$, where $T(\boldsymbol{x}; \Theta) = \sum_{j=1}^{J} \gamma_j \mathbb{1}\{\boldsymbol{x} \in R_j\}$. The policy tree itself does not produce decisions, but rather selects which candidate policy to invoke for a given input. Since all

policies $\pi^m$ produce feasible decisions, so does $\pi^{T(\boldsymbol{x};\Theta)}(\boldsymbol{x})$. The goal is to find the policy tree that solves the following minimization problem:

$$\hat{\Theta} \in \underset{\Theta}{\arg\min} \quad \mathbb{E}\left[c\big(\pi^{T(X;\Theta)}(X),Y\big)\right]. \tag{5}$$

By construction, the optimal policy tree cannot perform worse than the best single policy in the library. Moreover, if there exists a region of the covariate space where another policy offers a sufficiently large improvement in conditional expected cost, the optimal policy tree will achieve a corresponding improvement in overall performance. The following lemma formalizes this property.

**Lemma 1** *Suppose that $X, Y$ are obtained from a joint probability distribution, and let $m^\star$ be the index of the policy with the smallest overall expected cost $\mathbb{E}\left[c\big(\pi^m(X),Y\big)\right]$ from the library $\Pi_M = \{\pi^m\}_{m=1}^M$. Suppose there exists a region $R \subseteq \mathcal{X}$ with $\Pr(X \in R) > 0$ and a policy $m \neq m^\star$ such that $\mathbb{E}\left[c\big(\pi^m(X),Y\big)\,\middle|\,X \in R\right] \leq \mathbb{E}\left[c\big(\pi^{m^\star}(X),Y\big)\,\middle|\,X \in R\right] - \delta$, for some $\delta > 0$. Then $\mathbb{E}\left[c\big(\pi^{T(\boldsymbol{x};\hat{\Theta})}(X),Y\big)\right] \leq \mathbb{E}\left[c\big(\pi^{m^\star}(X),Y\big)\right] - \delta\cdot\Pr(X \in R)$.*

The proof follows because we can construct a tree that assigns policy $m$ to $R$ and policy $m^\star$ to its complement $R^c$. The expected cost of this tree is

$$\Pr(X \in R)\,\mathbb{E}\left[c\big(\pi^m(X),Y\big) \mid X \in R\right] + \Pr(X \in R^c)\,\mathbb{E}\left[c\big(\pi^{m^\star}(X),Y\big) \mid X \in R^c\right].$$

Subtracting the cost of always using $m^\star$ yields an improvement of at least $\delta \cdot \Pr(X \in R) > 0$. Since the optimal policy tree $T(.\,;\hat{\Theta})$ cannot do worse than this construction, the claim follows.

Although evaluating the expectation and handling the non-convex objective in equation 5 make computing $\hat{\Theta}$ challenging, several heuristic algorithms have been proposed to address this problem. In particular, we adopt the Optimal Trees algorithm for its strong empirical performance in similar policy selection tasks and its ability to scale to large datasets (Amram et al., 2022). This method uses a coordinate descent approach to minimize a regularized empirical risk version of equation 5. At each iteration, the current tree structure determines the best prescription for each leaf; these prescriptions are then evaluated in the objective, and the result guides the next coordinate descent step.

### 4.3  End-to-End Pipeline: Training and Inference Phases

We now integrate the *Prescribe* (Section 4.1) and *Select* (Section 4.2) components into a unified end-to-end framework *Prescribe-then-Select* (PS). The process consists of two stages: a training phase, which constructs both the candidate policies and a collection of policy trees for selecting the best among them, and an inference phase, which uses the ensemble of these trees to form a meta-policy that generates context-specific prescriptions.

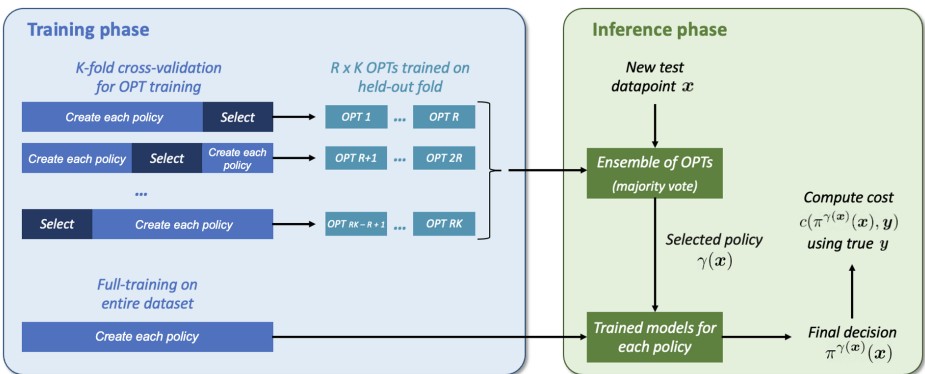

Figure 2: **End-to-end pipeline.** Left: $K$-fold, $R$-replicate training produces an ensemble of OPTs, each trained on a distinct held-out fold using its corresponding cost table. Right: At inference, a new context is routed through the OPT ensemble, and the majority-vote policy is applied to produce the final decision.

**Training phase.** Let $\mathcal{D}_{\text{train}} = \{(\boldsymbol{x}_i, \boldsymbol{y}_i)\}_{i=1}^N$ denote the training set, and let $\{\mathcal{I}^{(1)}, \ldots, \mathcal{I}^{(K)}\}$ be a random $K$-fold partition of the index set $[N]$. For a given fold $k$, the training indices $\mathcal{I}^{(-k)} := [N] \setminus \mathcal{I}^{(k)}$ are used to create the candidate policies $\pi^1, \ldots, \pi^M$ according to the procedures in Section 4.1. For each held-out instance $i \in \mathcal{I}^{(k)}$ and each candidate policy $m \in [M]$, we assess the quality of this policy on this observation via the realized out-of-sample cost: $C_{i,m}^{(k)} = c\left(\pi^m(\boldsymbol{x}_i), \boldsymbol{y}_i\right)$. Arranging these values for all $i \in \mathcal{I}^{(k)}$ yields the cost table $C^{(k)} \in \mathbb{R}^{|\mathcal{I}^{(k)}| \times M}$, where

$$C^{(k)} = \begin{bmatrix} c(\pi^1(\boldsymbol{x}_{i_1}), \boldsymbol{y}_{i_1}) & \cdots & c(\pi^M(\boldsymbol{x}_{i_1}), \boldsymbol{y}_{i_1}) \\ \vdots & \ddots & \vdots \\ c(\pi^1(\boldsymbol{x}_{i_{|\mathcal{I}^{(k)}|}}), \boldsymbol{y}_{i_{|\mathcal{I}^{(k)}|}}) & \cdots & c(\pi^M(\boldsymbol{x}_{i_{|\mathcal{I}^{(k)}|}}), \boldsymbol{y}_{i_{|\mathcal{I}^{(k)}|}}) \end{bmatrix}, \quad \{i_1, \ldots, i_{|\mathcal{I}^{(k)}|}\} = \mathcal{I}^{(k)}.$$

We use this k-fold cross-validation approach to prevent overfitting, since deploying the candidate policies out of sample prior to OPT training allows the OPT to identify which policies tend to generalize best in different regions of the covariate space. For each fold $k$, we train $R$ independent OPTs $T^{(k,1)}, \ldots, T^{(k,R)}$ which take as input the pairs $(\boldsymbol{x}_i, \mathbf{C}_{i,1:M}^{(k)})$ for $i \in \mathcal{I}^{(k)}$, solving equation 5 with $R$ different random seeds to mitigate the fact that the learning algorithm is a stochastic heuristic and may not always find a global optimum. After repeating this procedure for all $K$ folds, we have a total of $K \times R$ policy trees. Finally, all candidate policies $\pi^1, \ldots, \pi^M$ are refit on the full training set $\mathcal{D}_{\text{train}}$ to maximize the performance on the inference phase. A detailed, per-fold view of the procedure appears in Figure 3, while Algorithm 1 in the appendix summarizes the complete training phase.

**Inference phase.** Upon observing a covariate $\boldsymbol{x} \in \mathcal{X}$, each OPT $T^{(k,r)}$ outputs a policy index $\gamma^{(k,r)}(\boldsymbol{x}) \in [M]$, and we aggregate these outputs by majority vote $\gamma(\boldsymbol{x}) = \text{mode}(\{\gamma^{(k,r)}(\boldsymbol{x}) : k \in [K], r \in [R]\})$, breaking ties uniformly at random. The final decision made by the meta-policy is obtained by applying the selected policy $\pi^{\gamma(\boldsymbol{x})}(\boldsymbol{x})$ (see Figure 2). This procedure is explained in detail in Algorithm 2 in the appendix.

We highlight that PS does not use additional data beyond what each individual policy uses. All candidate policies and the meta-policy use the exact same data for model development and testing.

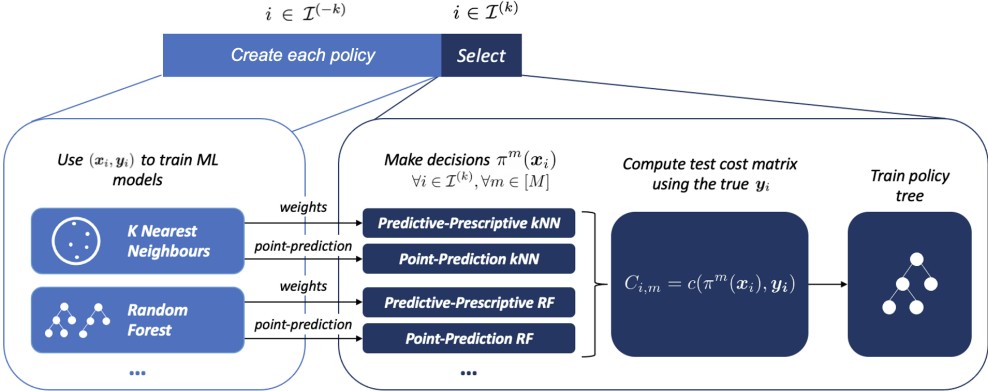

Figure 3: **Detailed view of one fold in the training phase.** In fold $k$, models trained on $\mathcal{I}^{(-k)}$ produce sample weights or point predictions, mapped to decisions $\pi^m(\boldsymbol{x}_i)$ for each $i \in \mathcal{I}^{(k)}$ and $m \in \Pi_M$. Evaluating these using the true outcomes $\boldsymbol{y}_i$ yields the cost matrix $C^{(k)}$, which, together with $\{\boldsymbol{x}_i : i \in \mathcal{I}^{(k)}\}$, trains $R$ Optimal Policy Trees. Across all folds, this produces $R \times K$ OPTs. We show an example with $M = 4$ policies (predictive–prescriptive kNN, point-prediction kNN, predictive–prescriptive RF, and point-prediction RF).

## 5 Computational Experiments

In this section we evaluate Prescribe-then-Select on two classical contextual stochastic optimization tasks that are widely used as benchmarks in the operations research literature (e.g. Bertsimas & Kallus (2020); Bertsimas et al. (2023); Huber et al. (2019); Ban & Rudin (2019); Bertsimas et al. (2018b)). The first task

is a two-stage shipment-planning problem, where initial production decisions are made at multiple facilities before demand is realized, followed by additional production and shipment decisions once demand becomes known. The second task is a single-stage newsvendor problem, a classic inventory management setting where stocking levels must be chosen in advance under uncertain demand. For both tasks, we introduce context heterogeneity by generating synthetic datasets with distinct demand segments—for example, holiday spikes versus routine weekday or weekend cycles—each following different distributions.

## 5.1 Multi–Product Newsvendor

The multi–product newsvendor problem is a classical model in inventory management (Khouja, 1999), where a decision-maker must determine order quantities for multiple products before uncertain demand is realized. It is widely used in retail, wholesale, and manufacturing to balance the costs of understocking (lost sales) and overstocking (excess holding or disposal). In the multi-product setting, items compete for a shared resource such as storage space or budget, and the decision-maker seeks to allocate this capacity as to maximize the expected profit. We consider the contextual version in which the decision-maker observes covariates as side information (e.g., calendar or seasonality indicators) before ordering.

Let $\boldsymbol{p}, \boldsymbol{c}, \boldsymbol{s} \in \mathbb{R}_+^d$ denote prices, costs, and storage requirements for $d$ products, which share a total storage capacity $S > 0$. The random vector $Y \in \mathbb{R}^d$ represents the (unknown) demand, and $\boldsymbol{x}$ the observed covariates. The decision maker chooses stocking levels $\boldsymbol{q}$ to maximize expected profit, formulated as

$$\max_{\boldsymbol{q} \geq 0} \ \mathbb{E}\big[\boldsymbol{p}^\top \min\{Y, \boldsymbol{q}\} - \boldsymbol{c}^\top \boldsymbol{q} \,\big|\, X = \boldsymbol{x}\big] \quad \text{s.t.} \quad \boldsymbol{s}^\top \boldsymbol{q} \leq S,$$

where the minimum $\min\{Y, \boldsymbol{q}\}$ is applied component-wise. In our experiments we set $d = 4$ and $S = 1200$. Prices $\boldsymbol{p}$, costs $\boldsymbol{c}$, and storage coefficients $\boldsymbol{s}$ are set such that higher-price items carry proportionally higher unit costs and larger storage requirements—while maintaining positive margins ($\boldsymbol{p} > \boldsymbol{c}$). This yields comparable profit magnitudes across products and nontrivial trade-offs under the knapsack constraint $\boldsymbol{s}^\top \boldsymbol{q} \leq S$. The exact parameter values for prices, costs, and storage coefficients are provided in Appendix B. Moreover, since $\mathbb{E}\big[\boldsymbol{p}^\top \min\{Y, \boldsymbol{q}\} \,\big|\, X = \boldsymbol{x}\big] = \sum_{i=1}^d p_i\, \mathbb{E}\big[\min\{Y_i, q_i\} \mid X = \boldsymbol{x}\big]$ (by linearity of the expectation), the implementation of both the PP and PPt policies for this problem estimate the distributions $f_{Y_j|X}(\cdot \mid \boldsymbol{x})$ for all $j \in [d]$ instead of $f_{Y|X}(\cdot \mid \boldsymbol{x})$.

## 5.2 Shipment Planning

Shipment–planning is a relevant problem to manufacturers, distributors, and logistics providers seeking to minimize transportation and production costs while meeting demand requirements (Bertsimas & Kallus, 2020). In a two–stage setting, the planner commits to an initial production plan before demand is observed, and then adjusts additional production and plans shipments once demand becomes known. We again assume the decision-maker has access to contextual information prior to making decisions.

Formally, $F$ production facilities, $L$ demand locations, and covariates $\boldsymbol{x} \in \mathcal{X}$ available as side information. In the first stage, before demand is observed, the planner chooses $\mathbf{u}_1 \in \mathbb{R}_+^F$ with unit cost $p_1$. After demand $Y \in \mathbb{R}_+^L$ is realized, additional items $\boldsymbol{e} \in \mathbb{R}_+^F$ at price $p_2 > p_1$ are produced, and shipments $u_{2,fl}$ from location $f$ to location $\ell$ at per–unit costs $c_{f\ell} \geq 0$ must be decided. Each product yields per-unit revenue $a > 0$ and full demand must be satisfied. The cost minimization CSO problem can then be written as

$$\min_{\mathbf{u}_1 \in \mathbb{R}_+^F} \ p_1\, \mathbf{1}_F^\top \mathbf{u}_1 \ + \ \mathbb{E}\big[Q(\mathbf{u}_1; Y) - a\, \mathbf{1}_L^\top Y \mid X = \boldsymbol{x}\big], \quad \text{where}$$

$$Q(\mathbf{u}_1; \boldsymbol{y}) = \min_{\mathbf{U}_2 \in \mathbb{R}_+^{F \times L},\, \boldsymbol{e} \in \mathbb{R}_+^F} \sum_{f=1}^F \sum_{\ell=1}^L c_{f\ell}\, u_{2,f\ell} + p_2\, \mathbf{1}_F^\top \boldsymbol{e} \quad \text{s.t.} \quad \mathbf{U}_2^\top \mathbf{1}_F \geq \boldsymbol{y}, \ \ \boldsymbol{e} \geq \mathbf{U}_2 \mathbf{1}_L - \mathbf{u}_1.$$

In our experiment, we set $F = L = 4$, $p_1 = 5$, $p_2 = 10$, $a = 90$, and draw fixed shipping costs via $c_{f\ell} = 20 + 2(f - 1) + \xi_{f\ell}$ with $\xi_{f\ell} \sim \text{Unif}[0, 3]$.

### 5.3 Implementation Details

*Data Generation:* Our data-generation process is designed to simulate demand data with explicit covariate-driven heterogeneity to create distinct demand segments, following common piecewise data-generation approaches (Amram et al., 2022; Zhou et al., 2023; Bertsimas et al., 2019). In the multi-product newsvendor setting, segments capture patterns such as holiday-driven spikes, smooth seasonal variation with weekday effects, and short-term disruptions. In the shipment-planning setting, segments reflect operational patterns such as contracted replenishment periods, event-driven surges, and routine demand with calendar effects. In both cases, regime activation is determined by calendar covariates, and we add gaussian noise to introduce variability across products or locations. Within each segment, demand is generated using smooth covariate-driven functions combined with i.i.d. Gaussian and Bernoulli distributions—consistent with standard practice in the literature (e.g. (Bertsimas & Kallus, 2020; Kallus & Mao, 2023; Bertsimas et al., 2023; Ban et al., 2019)). Full specifications, including functional forms, parameter values, and segment definitions, are provided in Appendix B and C. We highlight that each OPT in the PS framework learns its own partition directly from the training data, without access to the ground-truth segments used for data generation.

*Software Implementation:* Our implementation combines Python for the machine-learning components with Julia for the optimization layer. The main packages are Gurobi (v12.0.1) for mathematical programming, the OPT framework from Interpretable AI (v3.2.2) for model selection, and in Python, scikit-learn (v1.6.0) and pandas (v2.2.3). Gurobi is accessed through Julia, while preprocessing and evaluation run in Python.

*Implementation Setup:* We use $K = 5$ folds and $R = 10$ repetitions, yielding $K \times R = 50$ trees per ensemble. PS chooses among the base policies $\{\text{SAA}, \text{PP\textsc{t}–RF}, \text{PP–RF}, \text{PP\textsc{t}–kNN}, \text{PP–kNN}, \text{PP\textsc{t}–NN}\}$. We evaluate training sizes $N \in \{250, 500, 750, 1000, 1250, 1500, 2000, 3000, 5000\}$; for each $N$ we draw 100 independent training samples and assess strictly out-of-sample performance on a fixed test horizon. For the prediction models, we use kNN with 5 neighbors and RF with number of trees $B = 5$, and a feed-forward neural network with hidden layers of sizes $(16, 32, 16)$, ReLU activations, and early stopping.

*Evaluation:* We evaluate policies on data not used for training or meta-policy construction. Let $\mathcal{I}^{\text{test}}$ be the index set of held-out test points and $\mathcal{D}_{\text{test}} := \{(\boldsymbol{x}_i, \boldsymbol{y}_i) : i \in \mathcal{I}^{\text{test}}\}$. To evaluate a policy $\pi$, we compute its average test profit $cost(\pi) = \frac{1}{|\mathcal{I}^{\text{test}}|} \sum_{i \in \mathcal{I}^{\text{test}}} c(\pi(\boldsymbol{x}_i), \boldsymbol{y}_i)$ (notice that in the shipment-planning problem, evaluating $c(\pi(\boldsymbol{x}_i), \boldsymbol{y}_i)$ requires solving the second-stage optimization problem defining $Q(\pi(\boldsymbol{x}_i); \boldsymbol{y}_i)$. This guarantees feasibility of the second-stage decisions $\boldsymbol{U}_2$ with respect to the realized demand $\boldsymbol{y}_i$ and first-stage decision $\pi(\boldsymbol{x}_i)$, ensuring that all policy evaluations satisfy the full constraint structure of the two-stage model.) We report this quantity for the candidate policies $\{\pi^m\}_{m=1}^M$ as well as for the meta-policy $\pi^{\gamma(\cdot)}$. To quantify variability as a function of the training-set size $N$, we draw $S = 100$ independent training samples for each of the sizes considered and run the full pipeline in Section 4.3, yielding base policies $\{\pi_{N,s}^m\}_{m=1}^M$ and meta-policy $\pi_{N,s}^{\gamma(\cdot)}$ for $s \in [S]$. For policy $m$ in sample $s$ with data-size $N$, its average test cost is then $cost(\pi_{N,s}^m)$. We estimate the mean of $\{cost(\pi_{N,s}^m)\}_{s=1}^S$ over samples $s$ and construct a two-sided $100(1-\alpha)\%$ confidence interval (CI) using the Student-$t$ approximation:

$$\hat{\mu}_N^m = \frac{1}{S} \sum_{s=1}^S cost(\pi_{N,s}^m), \qquad \hat{\sigma}_N^m = \sqrt{\frac{1}{S-1} \sum_{s=1}^S \left(cost(\pi_{N,s}^m) - \hat{\mu}_N^m\right)^2}, \qquad \text{CI}_N^m = \hat{\mu}_N^m \pm t_{1-\alpha/2,\, S-1} \frac{\hat{\sigma}_N^m}{\sqrt{S}}.$$

### 5.4 Results

In this section, we report performance on the test set as described in Section 5.3 to address three main research questions. Unless stated otherwise, each curve in the plots shows mean profit across repeated samples with two-sided 95% $t$-based confidence intervals. We report profit as the negative of total cost for shipment planning and as revenue for the newsvendor problem, such that higher values are better in all figures.

#### 5.4.1 Segment-wise heterogeneity

**RQ1: Do the candidate CSO policies exhibit heterogeneous out-of-sample performance across the covariate space, or does a single policy dominate the rest?** To answer this question, we

compute out-of-sample performance as a function of training size for six candidate policies (SAA, PP–kNN, PPᴛ–kNN, PP–RF, PPᴛ–RF, and PPᴛ–NN) and three data regimes, and we analyze the results below.

**Newsvendor.** As shown in Figure 1, we indeed observe heterogeneity across segments:
- *Segment A (small variance for holiday-influenced products).* For small $N$, PPᴛ–RF attains the highest profit, consistent with a low-variance setting where point prediction works well. As $N$ grows, PP–RF matches PPᴛ–RF suggesting improvements in the local distribution estimates of the PP method.
- *Segment B (medium variance for products with seasonal smooth demand).* PP–kNN is best for small $N$, which is expected as the averaging effect tends to work well for continuous functions. As $N$ increases (e.g., 3000–5000), PP–RF becomes the best method.
- *Segment C (high-variance products with discontinuous summer demand).* PP–RF dominates for all training sizes. This is expected as kNN averaging effect fails to capture abrupt changes and point-prediction variants can be very inaccurate when the prediction falls on the wrong side of the demand discontinuity.

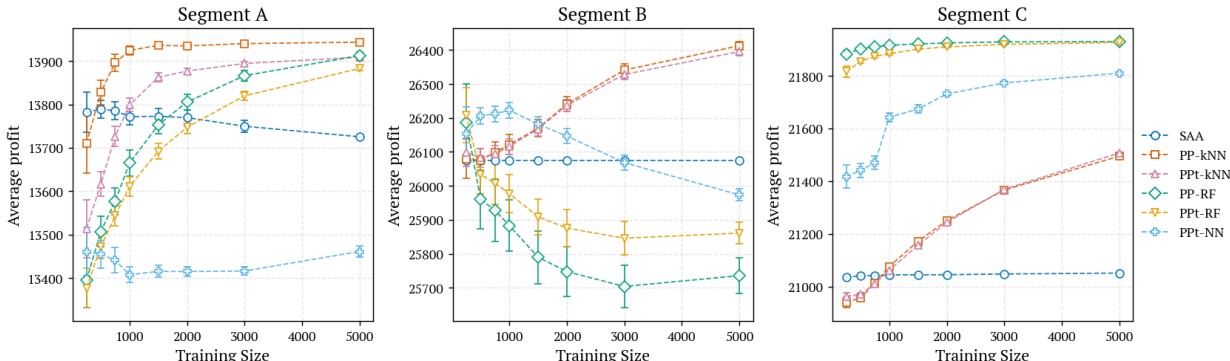

Figure 4: Shipment: segment-wise mean test profit vs. training size (95% CIs; profit $= -$ net cost).

**Shipment Planning.** Heterogeneity is also observed in this task, as shown in Figure 4:
- *Segments A/B (small-variance products/high-variance products with random discontinuities).* PP–kNN yields higher profit across most training sizes. Forests were observed to wrongly place calendar splits that affected performance in these segments, increasing leaf variance and harming the weights in the PP method; while kNN neighborhoods remained fairly stable.
- *Segment C (medium-variance for products with seasonal smooth demand).* PP–RF delivers the best performance, as the problem involves a complex continuous demand function. By contrast, kNN fails to capture the segment structure, lacking the flexibility of RF to adapt effectively. PPᴛ–NN attains intermediate performance: it improves with training size but remains below the RF methods for all $N$.

Across both tasks the answer to **RQ1** is affirmative: test performance is heterogeneous, and no single policy dominates uniformly. Which candidate policy is best depends on the segment and the training size. With limited data, the best policies differ by context (e.g., PPᴛ–RF in low-noise segments; PP–kNN where local averaging reduces variance), whereas in segments with abrupt regime changes PP–RF is consistently superior at all $N$. As $N$ grows, PP–RF closes the gap with point-prediction in simpler segments and usually overtakes other methods. We also observe cross-segment spillovers: because models are trained on pooled data, noise introduced by other segments (e.g., latent holiday/event effects) can make tree leaves noisier and depress RF performance even in otherwise regular segments, while kNN's locality is less sensitive to such noise.

### 5.4.2 Benefit of Prescribe-then-Select

**RQ2: Does Prescribe-then-Select outperform the best single policy, is the improvement statistically significant, and how does it vary with training size?** To answer this question, we conduct experiments on both benchmark problems, comparing PS against the best single-policy baseline across a range of training sizes. Figures 5a and 5b show average profit as a function of the training size with two-sided 95% $t$-based confidence intervals for PS and all candidate policies.

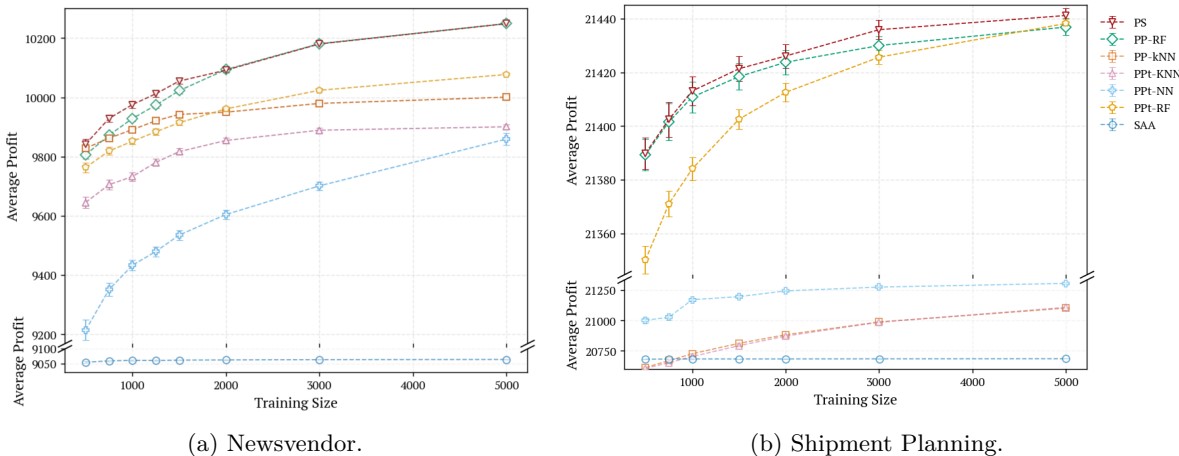

(a) Newsvendor.

(b) Shipment Planning.

Figure 5: Profit vs. training size for PS and the best single candidate policy in the (left) newsvendor and (right) shipment tasks. (95% CIs; profit $= -$ net cost)

**Newsvendor.** For small samples ($N \in [750, 1500]$), PS is statistically significantly better than all individual policies (non-overlapping or barely overlapping CIs at multiple $N$), reflecting its ability to route contexts toward PPT–RF (best in Segment A for small $N$) and PP–kNN (best in segment B for small $N$). The magnitude of the gain in this regime is modest in absolute terms but comparable to the improvement obtained when moving from point-prediction to predictive-prescriptive baselines. As data size increases ($N > 1500$), PP–RF dominates and PS rapidly converges to PP–RF (by $N \approx 2000$).

**Shipment Planning.** PS gains become apparent once the training set exceeds $N \approx 1000$, mirroring the pattern observed in 5.4.1: Segments A/B favor PP–kNN, whereas Segment C favors PP–RF in this data size regime. Improvements strengthen at $N \in \{3000, 5000\}$ where the 95% CIs only slightly overlap. Since the problem is two-stage with costly recourse ($p_2 \gg p_1$), choosing the wrong policy is highly expensive in terms of cost, which affects average gains even as PS increasingly routes A/B to PP–kNN and C to PP–RF.

Across both tasks, PS yields significant gains by learning partitions of the covariate space from cross-validation alone and finding the best candidate policy for each region. Notably in the shipment setting, policies that are very weak on average and would be disregarded in general frameworks—such as PP–kNN and PPT–kNN—become useful: PS deploys them in segments where they excel (A/B) and defaults to PP–RF elsewhere (e.g., C), thereby improving overall performance. Illustrative examples of the OPTs being trained are in the Appendix D. We note that some of the policies we consider—such as PP–kNN—are asymptotically optimal under standard assumptions: their expected loss converges to the optimal loss as the sample size grows (Bertsimas & Kallus, 2020). However, these experiments demonstrate that, with limited data sizes, no single policy from our library achieved the optimal loss during testing, highlighting the benefit of PS.

### 5.4.3 Uniform dominance and convergence

**RQ3: In the absence of substantial segment heterogeneity, does PS effectively revert to the best base policy, or does the selection step risk choosing an inferior policy?** In other words, we want to evaluate whether, in regimes where a single base policy is uniformly superior across the covariate space, PS asymptotically selects that policy and thus incurs negligible regret relative to the best single policy. In the newsvendor task (Figure 5a), for $N \gtrsim 2000$ the mean profits of PS and PP–RF are statistically indistinguishable within two-sided 95% $t$-based confidence intervals, indicating effective convergence of PS to the dominant policy. Because PS is trained solely via cross-validated folds and never observes test data, this demonstrates PS is a low-risk default: when uniform dominance emerges, PS matches the best single policy, while at heterogeneous cases it retains the performance gains.

### 5.4.4 Summary of findings and computational considerations

Our results show that (i) out-of-sample performance is segment-dependent, with no single candidate policy dominating the others; (ii) PS delivers statistically significant improvements over all single policies in het-

erogeneous regimes; and (iii) when a single base policy becomes uniformly dominant, PS converges to it.

**Computational Cost.** With $K$ cross-validation folds and $M$ candidate policies, each ML model is trained $K$ times, and is evaluated for all $M$ variants on each fold—yielding $K \times M$ sets of optimization problems to solve. Even though creating the meta-policy introduces an additional step, its cost is small compared to solving the optimization models themselves (see Appendix E). Thus, policy selection adds minimal overhead relative to standard CSO workflows where multiple policies are typically evaluated to identify the most suitable one. The only additional cost specific to PS is training the meta-policy, an ensemble of $K \times R$ OPTs, which adds modest training time and memory overhead and incurs negligible inference latency.

## 6 Conclusion

We introduced Prescribe-then-Select, a modular framework for adaptive policy selection in contextual stochastic optimization. PS generates a diverse set of candidate policies and learns a selection model that matches each context to its most effective policy, enabling flexible integration of diverse policies while preserving constraint feasibility. Our experiments on two benchmark problems showed that PS consistently exploits heterogeneity in the covariate space to improve upon the best single policy, while converging to the dominant policy in homogeneous regimes. These findings position PS as a practical, low-risk approach for decision-making environments where diverse, high-quality prescriptive models are already available. Future work includes extending PS to multi-stage settings, exploring other selection models beyond policy trees, and evaluating performance in large-scale, real-world applications with high-dimensional and partially observed covariates. An additional direction is to study cases where policy averaging is feasible and to consider the development of meta-policies that learn how to combine candidate policies instead of selecting only one.

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
