## A   Algorithms

---

**Algorithm 1: Training phase:** fold-wise construction of cost matrices and training of an OPT ensemble. Each $C^{(k)}$ contains realized out-of-sample costs from prescriptions $\pi^m(\boldsymbol{x}_i)$ (fit on $\mathcal{I}^{(-k)}$) and true outcomes $\boldsymbol{y}_i$ for $i \in \mathcal{I}^{(k)}$.

---

**Data:** $K$-fold partition $\{\mathcal{I}^{(1)}, \ldots, \mathcal{I}^{(K)}\}$ of $[N]$;   candidate policies $\{\pi^m\}_{m=1}^M$;   repetitions $R$;   OPT
          hyper-parameters $(D_{\max}, n_{\min}, \lambda)$
**Result:** Ensemble of OPTs $\{T^{(k,r)}\}_{k \in [K],\ r \in [R]}$;   refit policies $\{\pi^m\}_{m=1}^M$
**for** $k = 1$ **to** $K$ **do**
  **for** $m = 1$ **to** $M$ **do**
    Fit $\pi^m$ using $\{(\boldsymbol{x}_i, \boldsymbol{y}_i) : i \in \mathcal{I}^{(-k)}\}$ (incl. internal tuning)          *// fit only on in-fold complement*
  Initialize $C^{(k)} \in \mathbb{R}^{|\mathcal{I}^{(k)}| \times M}$          *// held-out cost matrix*
  **foreach** $i \in \mathcal{I}^{(k)}$ **do**
    **for** $m = 1$ **to** $M$ **do**
      $C_{i,m}^{(k)} \leftarrow c(\pi^m(\boldsymbol{x}_i),\ \boldsymbol{y}_i)$          *// evaluate using true $\boldsymbol{y}_i$*
  **for** $r = 1$ **to** $R$ **do**
    $T^{(k,r)} \leftarrow \text{TrainOPT}\Big(\{(\boldsymbol{x}_i, \mathbf{C}_{i,1:M}^{(k)}) : i \in \mathcal{I}^{(k)}\};\ D_{\max}, n_{\min}, \lambda,\ \text{seed} = r\Big)$   *// train with distinct seeds*
**for** $m = 1$ **to** $M$ **do**
  Refit $\pi^m$ on $\mathcal{D}_{\text{train}}$          *// final refit for deployment*

---

**Algorithm 2: Decision phase:** majority-vote selection via $\gamma^{(k,r)}(\boldsymbol{x})$, followed by applying the chosen refit policy.

---

**Data:** New context $\boldsymbol{x}$;   OPT ensemble $\{T^{(k,r)}\}_{k=1..K,\ r=1..R}$;   refit policies $\{\pi^m\}_{m=1}^M$
**Result:** Prescription $\boldsymbol{z}$
**for** $k = 1$ **to** $K$ **do**
  **for** $r = 1$ **to** $R$ **do**
    $\gamma^{(k,r)}(\boldsymbol{x}) \leftarrow T^{(k,r)}(\boldsymbol{x})$  *// policy index in $[M]$*
$\gamma(\boldsymbol{x}) \leftarrow \text{mode}\big(\{\gamma^{(k,r)}(\boldsymbol{x}) :\ k \in [K],\ r \in [R]\}\big)$
**if** *tie* **then**
  break uniformly at random
**return** $\pi^{\gamma(\boldsymbol{x})}(\boldsymbol{x})$

---

# B    Multi-Product Newsvendor: Experimental Details

## B.1    Data Generation

We simulate demands for $d$ products over time $t$ using the following calendar covariates: day of week $dow_t \in \{0, 1, \ldots, 6\}$, day of month $dom_t \in [31]$, month $M_t \in [12]$, day of year $doy_t \in [366]$, weekend indicator $\omega_t = \mathbb{1}\{dow_t \geq 5\}$, and holiday indicator $h_t \sim \text{Bernoulli}(p_{\text{hol}})$ with $p_{\text{hol}} = 0.1$. We define three covariate regimes: Segment A models holiday–sensitive products (e.g., gifts); Segment B features smooth seasonality with weekday modulation; Segment C introduces abrupt midsummer weekday jumps, representing short promotions or disruptions. In Segment C, $s_{M_t}$ is a month–specific offset: $s_7 = -7$ (July) and $s_8 = +8$ (August), producing discontinuous changes without trend.

Realized demands on day $t$ are $Y_{jt} = \max\{0, \mu_{jt} + \varepsilon_{jt}\}$, with noise $\varepsilon_{jt} \sim \mathcal{N}(0, \sigma_A^2)A_{jt} + \mathcal{N}(0, \sigma_B^2)B_{jt} + \mathcal{N}(0, \sigma_C^2)C_t$, where $A_{jt}$, $B_{jt}$, and $C_t$ are binary indicators for product $j$ at time $t$ being in Segment A, Segment B, or Segment C, respectively. Noise is independent across products on any given day and segment. The specification for each regime is summarized in Table 1.

| | Segment A (holiday–sensitive) | Segment B (seasonal/weekday) | Segment C (summer jump) |
|---|---|---|---|
| Activation | $A_{jt} = \mathbb{1}\{h_t = 1, j \in \{0, 1\}\}$ | $B_{jt} = 1 - \max(A_{jt}, C_t)$ | $C_t = \mathbb{1}\{M_t \in \{7, 8\}, dow_t \leq 3\}$ |
| Business rationale | Holiday–driven lift for gift-suitable items | Seasonal cycle with weekday variation | Short promotions or disruptions in midsummer |
| Qualitative pattern | Sharp, low–variance holiday spikes | Smooth annual wave modulated by weekdays | Large weekday jumps in July/August |
| Mean $\mu_{jt}$ | $B + \alpha_j$ | $B + 6\sin\frac{2\pi M_t}{12} \cdot \frac{dow_t + 1}{5} \cdot (1 + 0.15\,j)$ | $B + s_{M_t} + 4j$ |
| Noise scale | $\sigma_A = 0.5$ | $\sigma_B = 3.0$ | $\sigma_C = 4.0$ |

Table 1: Segment specification for multi–product newsvendor demand. Mean $\mu_{jt}$ and noise scale depend on the active segment. Baseline $B = 30$, holiday lifts $\alpha_0 = 8$, $\alpha_1 = 5$, and step adjustments $s_{M_t} \in \{-7, +8\}$ for $M_t \in \{7, 8\}$.

## B.2    Parameters

Table 2 reports the selling prices $p_i$, procurement costs $c_i$, and storage coefficients $s_i$ used in our experiments.

Table 2: Selling prices, procurement costs, and storage coefficients for the products used in the multi–product newsvendor experiments.

| Product | Price ($p_i$) | Cost ($c_i$) | Storage ($s_i$) |
|---|---|---|---|
| Product 0 | 500.0 | 350.0 | 3.0 |
| Product 1 | 800.0 | 600.0 | 15.0 |
| Product 2 | 50.0 | 30.0 | 1.5 |
| Product 3 | 10.0 | 6.0 | 0.5 |

## C Shipment Planning: Experimental Details

### C.1 Data Generation

For the shipment–planning setting, we create demand regimes distinct from the multi–product newsvendor case, while maintaining realistic patterns. This yields a complementary experiment with different sources of heterogeneity. We simulate demands for $l$ locations using the same calendar covariates $(dow_t, dom_t, M_t, \mathrm{doy}_t, \omega_t, h_t, H_t)$, and latent driver $H_t \sim \mathcal{N}(0, 10^2)$, that is not included as feature in the model. Exactly one of three segments (A/B/C) is active per day, summarized in Table 3.

Each location $\ell$ has offset $\delta_\ell = \sin\left(\frac{2\pi(\ell-1)}{L}\right)$, and realized demands are $Y_{\ell t} = \max\{0, \mu_t + \delta_\ell + \varepsilon_{\ell t}\}$, with independent noise $\varepsilon_{\ell t} \sim \mathcal{N}(0, \sigma_A^2)A_t + \mathcal{N}(0, \sigma_B^2)B_t + \mathcal{N}(0, \sigma_C^2)C_t$.

|  | Segment A (early–month) | Segment B (holiday/event) | Segment C (routine) |
|---|---|---|---|
| Activation | $A_t = \mathbb{1}\{dom_t \le 8,\, M_t \le 4\}$ | $B_t = \mathbb{1}\{h_t = 1\}$ | $C_t = 1 - \max(A_t, B_t)$ |
| Business rationale | Contracted early–month replenishment | Event–driven surges | Regular operations |
| Qualitative pattern | Flat mean, low variance | Short, high–variance spikes | Gradual trend with weekday/weekend shifts |
| Mean $\mu_t$ | $B + 25$ | $B + 5 + 20\,H_t$ | $B + 0.08\sqrt{\mathrm{doy}_t} + 4(dow_t)^2 + 10\,\omega_t$ |
| Noise scale | $\sigma_A = 0.3$ | $\sigma_B = 4.0$ | $\sigma_C = 1.2$ |

Table 3: Segment specification for shipment demand. One segment is active per day $t$; $\mu_t$ and noise scale depend on the active segment. Baseline $B = 30$.

# D    Illustrative Trees from Ensemble

In this section we provide illustrative examples of the OPTs learned by the PS framework. In Figures 6 and 7 we observe that the partitions discovered by the OPTs did not perfectly recover the underlying segments used for data generation, but we did observe meaningful overlap: some parts of the OPT partition aligned with the ground-truth segments, suggesting that OPT identified heterogeneity in policy performance across those segments.

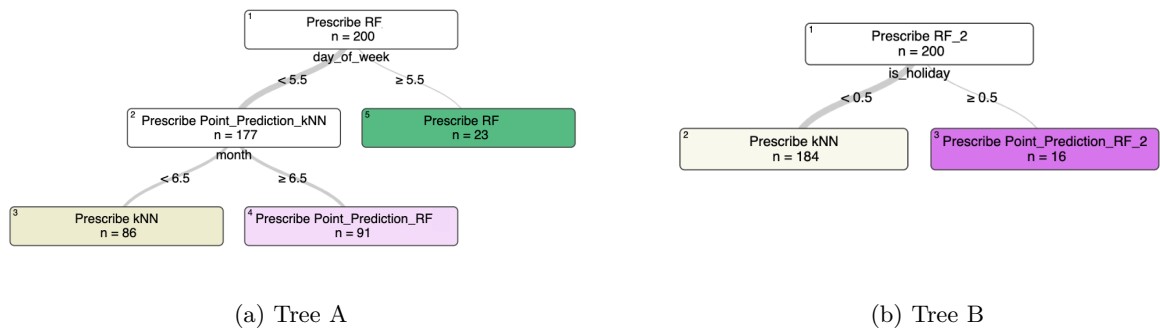

(a) Tree A                                                        (b) Tree B

Figure 6: Illustrative OPTs for the multi–product newsvendor. Two randomly chosen trees from the PS ensemble ($K = 5$ folds, $R = 10$ repetitions, $K \times R = 50$ trees) on one sample with $N = 1000$. **Left**: the tree approximates segment C (summer jumps) using `day_of_week` $< 5.5$ (weekdays) and `month` $\geq 7$ (July–August); it prescribes PPt–RF in that region which is the second best model. For the remaining cases with `day_of_week` $< 5.5$, it routes to kNN, which is best for segment B (most of the remaining of the data). **Right**: the tree partially isolates segment A by splitting on `is_holiday` and prescribing PPt–RF on holidays, which is the best for segment A; non-holiday days go to kNN, which is the best in the bigger segment B. Each tree is trained on a cross-validation fold and is imperfect on its own, but the ensemble (majority vote over 50 trees) aggregates these partial signals into an effective meta-policy.

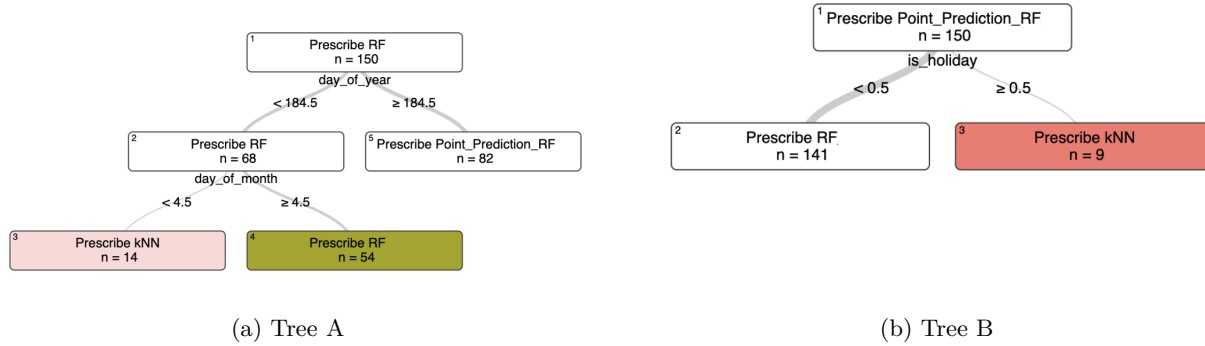

(a) Tree A                                                        (b) Tree B

Figure 7: Illustrative OPTs for the shipment–planning task. Two randomly chosen trees from the PS ensemble ($K = 5$ folds, $R = 10$ repetitions, $K \times R = 50$ trees) on a sample with $N = 3000$. **Left**: the tree uses `day_of_year` $\leq 184.5$ (roughly first half of the year) and `day_of_month` $< 4.5$ (very early month) to approximate segment A, prescribing PP–kNN in that region (the best for segment A). For the remaining contexts it prescribes PP–RF or PPt–RF, consistent with segment C (routine), but it does not isolate segment B. **Right**: the tree splits on `is_holiday`, cleanly isolating segment B (holiday/event) and prescribing PP–kNN there (the best for segment B), while routing non-holiday days to RF, which aligns with the bigger segment C. Each tree is trained on a cross-validation fold and is imperfect on its own; the ensemble (majority vote over 50 trees) aggregates these partial signals to recover an approximation to the segment structure.

# E Runtime Analysis

In Table 4 we report mean wall–clock seconds for the training and inference phases (averaged across 10 randomly sampled training datasets) for each training size $N$. The columns correspond to the time it takes to perform (1) data preprocessing (e.g. splitting and encoding the data); (2) ML model training on the entire dataset (RF, kNN, NN); (3) extracting weights for the PP approach (RF, kNN) as well as predictions for the PPt approach; (4) cross–validation across folds (per–fold ML model training, extracting weights and predictions, and computing validation results); (5) solving all optimization problems across all policies and testing data; and (6) training the Optimal Policy Tree ensemble and running the inference phase on the testing set. Across both tasks, the dominant computational cost remains solving the underlying optimization problems. Even though the meta-policy introduces an additional step, its cost is small compared to solving the optimization models themselves. For example, at $N$=5000, the newsvendor benchmark takes 1064.69 s in optimization vs. 31.21 s in the meta–policy; shipment spends 390.66 s vs. 18.87 s. Thus, policy selection adds minimal overhead relative to standard CSO workflows where multiple policies are evaluated to identify the most suitable one.

Lastly, we note that memory usage is low throughout our experiments, with peak consumption remaining below 0.5 GB even for the largest training sizes.

Table 4: Average wall–clock runtime (seconds) across 10 random training sets — Multi–Product Newsvendor.

| Training size | Data Prep (Python) | ML Models Train-on-Full (Python) | Neighbor Tables (Python) | ML Models Cross-Validation (Python) | Optimization Models (Julia) | Meta-Policy (Julia) |
|---|---|---|---|---|---|---|
| 250 | 0.01 | 0.28 | 1.66 | 1.59 | 52.29 | 16.48 |
| 500 | 0.01 | 0.39 | 1.70 | 2.44 | 77.30 | 14.74 |
| 750 | 0.01 | 0.61 | 1.71 | 3.39 | 102.86 | 13.59 |
| 1000 | 0.01 | 0.74 | 1.72 | 4.37 | 184.49 | 15.18 |
| 1500 | 0.01 | 1.04 | 1.72 | 6.16 | 177.70 | 19.51 |
| 2000 | 0.01 | 1.29 | 1.74 | 7.94 | 252.55 | 31.89 |
| 3000 | 0.01 | 1.89 | 1.78 | 11.31 | 430.05 | 29.33 |
| 5000 | 0.01 | 3.28 | 2.10 | 20.25 | 1064.69 | 31.21 |

Table 5: Average wall–clock runtime (seconds) across 10 random training sets— Shipment Planning.

| Training size | Data Prep (Python) | ML Models Train-on-Full (Python) | Neighbor Tables (Python) | ML Models Cross-Validation (Python) | Optimization Models (Julia) | Meta-Policy (Julia) |
|---|---|---|---|---|---|---|
| 250 | 0.02 | 0.22 | 0.25 | 1.20 | 48.25 | 11.89 |
| 500 | 0.02 | 0.25 | 0.25 | 1.27 | 60.51 | 10.74 |
| 750 | 0.02 | 0.25 | 0.25 | 1.40 | 72.18 | 11.33 |
| 1000 | 0.02 | 0.28 | 0.25 | 1.50 | 87.35 | 15.34 |
| 1500 | 0.02 | 0.35 | 0.25 | 2.06 | 105.61 | 15.71 |
| 2000 | 0.02 | 0.42 | 0.24 | 2.26 | 136.28 | 17.53 |
| 3000 | 0.02 | 0.55 | 0.24 | 3.04 | 203.77 | 16.34 |
| 5000 | 0.02 | 0.80 | 0.26 | 4.49 | 390.66 | 18.87 |

## F  Optimal Policy Trees for the Meta-Policy

As described in Amram et al. (2022), optimal policy trees solve a treatment assignment problem from observational data. The OPT method learns a decision tree that maps covariates to the treatment that optimizes expected outcomes. Importantly, we do not use the OPT to obtain a feasible policy for the CSO problems, since just like parametric decision rules; the OPT could not be tractably optimized to satisfy hard constraints on the decisions. We instead leverage OPTs to select among a set of feasible candidate policies based on the observed covariates. In other words, we consider each candidate policy as a treatment; and train the OPTs to learn which treatment is best given the contextual information. This use of OPT is novel in the sense that the "treatment" OPT selects among are entire optimization pipelines (e.g. PP-kNN; PP-RF), each solving a constrained optimization problem—not the primitive scalar/multiclass treatments OPT is commonly applied to.

We first restate the policy selection problem with the notation used in the main text. Let the candidate-library be $\Pi_M = \{\pi^m\}_{m=1}^M$ and let $T(\boldsymbol{x}; \Theta)$ be a depth-constrained, axis-aligned decision tree that partitions $\mathcal{X}$ into disjoint regions $\{R_j\}_{j=1}^J$ and assigns a policy index $\gamma_j \in [M]$ to each region. For any context $\boldsymbol{x}$, the meta-policy outputs the index $T(\boldsymbol{x}; \Theta) = \sum_{j=1}^J \gamma_j \mathbb{1}\{\boldsymbol{x} \in R_j\}$ and deploys $\pi^{T(\boldsymbol{x};\Theta)}(\boldsymbol{x})$, which is feasible because every $\pi^m$ is feasible by construction.

**Empirical objective on a fold.**  Recall that in fold $k$ we build the cost table $C^{(k)} \in \mathbb{R}^{|\mathcal{I}^{(k)}| \times M}$ with entries $C_{i,m}^{(k)} = c(\pi^m(\boldsymbol{x}_i), \boldsymbol{y}_i)$ for $i \in \mathcal{I}^{(k)}$ and $m \in [M]$. The policy-selection objective in equation 5 is approximated by its regularized empirical analogue on the held-out fold:

$$\hat{\Theta}^{(k)} \in \arg\min_{\Theta} \frac{1}{|\mathcal{I}^{(k)}|} \sum_{i \in \mathcal{I}^{(k)}} C_{i, T(\boldsymbol{x}_i;\Theta)}^{(k)} + \lambda \operatorname{splits}(T) \quad \text{s.t.} \quad \operatorname{depth}(T) \leq D_{\max}, \; |R_j| \geq n_{\min} \; \forall j. \quad (6)$$

Here, $\operatorname{splits}(T)$ is the number of internal nodes (complexity penalty), $D_{\max}$ bounds the depth of the tree, and $n_{\min}$ enforces minimum leaf size.

**Leafwise optimal assignments.**  The optimization problem in equation 6 is separable across leaves once the partition is fixed. If $\{R_j\}_{j=1}^J$ is given, the optimal policy index in leaf $j$ is simply assigned as

$$\gamma_j^\star \in \arg\min_{m \in [M]} \sum_{i \in \mathcal{I}^{(k)}: \, \boldsymbol{x}_i \in R_j} C_{i,m}^{(k)}. \quad (7)$$

This closed form is what makes the coordinate-descent training used by the Optimal Trees framework possible: one alternates between (i) updating the tree structure (splits) to improve the regularized empirical objective equation 6, and (ii) recomputing the leaf assignments via equation 7; see Amram et al. (2022) for details.

**From per-fold training to the ensemble.**  We train $R$ trees per fold, $\{T^{(k,r)}\}_{r=1}^R$, to mitigate heuristic stochasticity; after all $K$ folds are processed we update each candidate policy $\{\pi^m\}_{m=1}^M$ to use the entire training set. At inference, a new $\boldsymbol{x}$ is routed through all $K \times R$ trees and the final policy index is chosen by majority vote $\operatorname{mode}\{\gamma^{(k,r)}(\boldsymbol{x})\}$, after which the corresponding policy is applied (Algorithm 2).