# OpenReview forum: "Prescribe-then-Select: Adaptive Policy Selection for Contextual Stochastic Optimization"
_TMLR — Accepted by TMLR_

### Review · Reviewer_ChFf · 2025-10-24

**Summary Of Contributions:**

This paper considers contextual stochastic optimization with covariate x, decision variable z, random noise y, and objective function of the form $\min_{z} E[c(z, y)|x]$. This paper focuses on learning a meta policy to select among a set of candidate policies $\pi(z|x)$ given the covariate x. This process is called prescribing (candidate policies) and selecting (meta policy). The meta policy is represented as a policy tree. Using the newsvender and shipment planning problems, the authors demonstrate that the meta policy effectively selects the best policy for any covariate space partition when no single candidate policy dominates all partitions.

**Audience:**

Yes

**Audience Explanation:**

Contextual stochastic optimization is a topic of interest to the optimization and control community. The proposed method, which is essentially a two-staged learning of mixture-of-experts policy, is potentially of interest to some.

**Claims And Evidence:**

No

**Claims Explanation:**

The results show that the proposed prescribe-then-select method is effective, which is the mainly claim of the paper. Description of the methods is relatively clear. But some of the design choices seem a bit arbitrary.

**Requested Changes:**

My main comment is that some of the design choices in the paper seem a bit arbitrary.
* In section 4.3, you referred to the process described in section 4.1 as training policies. But what you are actually training, i.e., the KNN and RF, are predictive models of the form $P(y|x)$ or its simplification. Can you confirm this is the case, i.e., you don't directly train a mapping $\pi(z|x)$ and at inference time you also solve $\min_{z}$ rather than querying $\pi(z|x)$?
* Why did you choose KNN and RF as your predictive model classes? What about other models like generalized linear models, Gaussian process, or neural networks?
* Why is it fundamentally not possible (or considered) to train a globally optimal policy? With the amount of work or compute used by your method, you can presumably improve your probabilistic predictive model such that the resulting policy is better globally then the meta policy or any individual policy in your set?
* Can you explain more clearly the notion of partitioning by the policy tree, segments in the experiments section, and their relationship? I thought the covariate partitioning is supposed to be discovered automatically by the policy tree. Is that how you identified the segments or the identified partitions somehow overlapped with ground truth segments?

---

### Review · Reviewer_jKrC · 2025-10-24

**Summary Of Contributions:**

The paper presents a method that learns to choose the best Contextual Stochastic Optimization policy from a set of candidate policies, depending on the observed covariates. This is enabled by first training candidate policies on data representative of the entire covariate space, and then learning a policy-labeled partition of the covariate space. Given trained policies, the policy-labeled partition of the covariate space is realized via the Optimal Policy Trees (OPTs) framework. Given a covariate, the an ensemble of OPTs then routes this query to the respective candidate policy.

The authors show theoretically that the optimal policy is never worse than the best single policy, which is validated empirically on two tasks.

**Audience:**

Yes

**Audience Explanation:**

Finding the optimal policy in CSO is a relevant problem and the paper provides a useful contribution.

**Broader Impact Concerns:**

This is fundamental research, there are no immediate ethical implications.

**Claims And Evidence:**

Yes

**Claims Explanation:**

**Strengths**
- The need for the method is clearly motivated (Section 5.4.1).
- The method is explained in a clear manner, the exposition is easy to follow.
- The theoretical result (Lemma 1) is sensible and the empirical evaluation supports the findings nicely.

**Weaknesses**
- The method comes with a significantly larger training cost, as $M \times K + M$ individual policy and $R \times K$ policy tree training routines are required. Whether this is prohibitive depends on the training resources required for the individual policies.

**Questions and Comments**

1. Why is the k-fold CV method chosen to train the OPTs? Specifically, why is each OPTs only fit to the respective held out data, instead of the entire data set? Was this tried and overfitting was observed? Why is it necessary to deploy the candidate policies out of sample in the OTP fitting procedure?

2. Section 5.4.1 provides a nice motivation for why a single policy is insufficient. Perhaps a teaser for this, e.g., in form of a figure, could be shown earlier in the paper.

3. Are all relevant candidate policies considered in this work? Is there truly not a single policy that can capture $f_{Y \mid X}(y \mid x)$, thought the covariate space? This would be relevant to determine the significance of this work.

4. Final paragraph of Section 4: These [...] $\rightarrow$ This [...]

**Requested Changes:**

See above.

---

### Review · Reviewer_e4ii · 2025-11-12

**Summary Of Contributions:**

## Summary

This paper studies the contextual stochastic optimization (CSO) problem,
where upon receipt of side information, a decision from among a
continuous set must be chosen to minimize a (stochastic) cost. Towards
this end, the authors introduce Prescribe-then-Select (PS), a framework
for combining a set of "base" decision policy learning algorithms to
produce a superior policy that performs uniformly well across the
context space. The approach uses an ensemble of Optimal Policy Trees
(OPTs)—a decision tree method over policies—together with an majority
vote to determine which base policy to deploy for any given context.
Through experiments on synthetic datasets, the paper shows that PS
indeed outperforms all base policies.

## Strengths

1.  The paper, for the most part, is very nicely written and a joy to
    read. The authors took great care to make every statement precise,
    which I greatly appreciate.
2.  Similarly, the implementation details are given in very nice detail;
    even the submitted code is very well documented.
3.  The paper states its research questions / hypotheses very clearly,
    and gives precise conclusions in support of their hypotheses.
4.  While not a result about the proposed algorithm, I really
    appreciated the added touch of RQ1 and the corresponding
    experimental results, which demonstrates that existing methods do
    indeed exhibit highly context-dependent performance (and the ranking
    of these methods fluctuates across context space).

## Weaknesses

1.  The paper says PS constructs a "diverse library of feasible
    candidate policies", but this feels like a stretch to me. The
    candidate policies are all very simple; one of them is
    context-independent, and the remaining 4 are actually just two
    distinct methods instantiated across two function approximators (kNN
    and random forest).
2.  The paper also claims to conduct "extensive computational
    experiments" on benchmark CSO problems. Again, this feels like a
    stretch. Firstly, only two benchmarks are tested. This isn't a huge
    deal in my opinion because the two benchmarks have quite different
    properties. But also, these benchmarks are synthetic, and I
    personally can't appreciate how relevant they are as someone that is
    not involved in CSO / operations management research. That said,
    "extensive computational experiments" lead me to expect experiments
    on real datasets.
3.  Despite the very clear writing and the otherwise nice related work
    section, it is not actually clear to me what the main innovation of
    this paper is, given that OPT already exists.

**Additional Comments:**

Below are some comments I had while reading the paper; mostly just
elaborating a little more on some of the requested changes.

In section 4.1, how does one typically perform the optimization over
$\mathcal{Z}$, which is a continuous space? Is there an assumption that
$c$ is convex in the first argument? Particularly with regard to PPt, I
would expect that $c$ should be convex in both arguments.

It would help if you could explain more what the difference is between
your method and OPT. From my understanding, PS is OPT applied directly
to CSO. What were the challenges involved in this application? Why don't
the results from OPT trivially carry over to this setting?

My hunch is that the major difference comes from the 2-phase approach,
where you build an ensemble of OPT trees in the first phase. But this
concept doesn't seem specific to CSO to me. Could you have used the same
technique in the exact setting of the OPT paper to achieve better
performance?

Is there any reason why you can't use e.g. neural network policies
within PS? Is it just due to difficulties of realizing feasible
policies? If that's the case, are there any meaningful benchmarks you
can run where it is easy to represent feasible policies with neural
networks / other function approximators?

**Audience:**

Yes

**Audience Explanation:**

The CSO problem is quite relevant to the ML research community. The
success of fixing context-dependent performance via a meta-policy will
likely be interesting to the TMLR audience.

**Broader Impact Concerns:**

No broader impact concerns.

**Claims And Evidence:**

Yes

**Claims Explanation:**

The theoretical claim appears to be correct, albeit simple. The
empirical claims appear to be correct as well, and are supported by
many appropriate experimental results.

**Requested Changes:**

1.  **Critical**. Currently, the policy library consists of very simple
    policies based on classical ML techniques. I would be really curious
    to see if the influence of PS is as major when the library consists
    of more modern policy instantiations, such as deep neural nets. If
    this is infeasible, it should at least be discussed. I mention
    neural networks primarily because these are useful tools for
    ingesting contexts and implementing complex context-dependent
    decisions, so perhaps a single policy of this nature can render PS
    obsolete.
2.  **Critical**. I believe the benchmarks need further justification
    here. At the very least, the method of sampling the synthetic data
    should be discussed / motivated more. Are these synthetic datasets
    standard? Does the CSO / operations management community understand
    / care about these benchmarks? Ideally, experiments on a real
    dataset would be very nice.
3.  **Critical**. Related to the first point, I believe some important
    baseline methods are missing here. Many are mentioned in the related
    work, but I don't think any of those are tested. I think again this
    is especially relevant since stronger baselines could be
    incorporated into PS, and I'd like to see that PS is still useful
    even with stronger base policies.
4.  **Non-critical**. Please discuss OPT in more detail, particularly to
    elucidate how your approach is an extension of it.

---

### Review · Reviewer_xgRy · 2025-11-12

**Summary Of Contributions:**

This paper proposes Prescribe‑then‑Select (PS), a modular framework for contextual stochastic optimization that first builds a library of feasible candidate policies (e.g., SAA, point‑prediction and predictive‑prescriptive variants with kNN and Random Forests) and then trains a meta‑policy (ensembles of Optimal Policy Trees) to choose, per context, which candidate to deploy. Using synthetic shipment‑planning and multi‑product newsvendor benchmarks with deliberately heterogeneous demand regimes, PS improves over any single policy when heterogeneity is present and effectively collapses to the dominant policy when one exists.

**Audience:**

Yes

**Audience Explanation:**

The work sits at the interface of learning‑to‑decide and robust prescriptive analytics, reusing strong prescriptive models while adding a learned gating layer that respects feasibility. This modularity is attractive to practitioners who already have trusted policies but face context heterogeneity. It complements approaches such as “predict‑then‑optimize” by learning when each paradigm works rather than committing to one.

**Broader Impact Concerns:**

In high‑stakes domains (e.g., healthcare logistics, loan default predictions), context‑conditioned selection can encode spurious correlations. Mis‑gating could degrade service for minority contexts.

**Claims And Evidence:**

Yes

**Claims Explanation:**

- The paper’s central claims that (i) candidate CSO policies perform heterogeneously across contexts, (ii) a learned selector can improve over any single policy in such regimes while (iii) reverting to the dominant policy when heterogeneity fades, are supported by clear experimental evidence on two standard CSO benchmarks (multi‑product newsvendor and shipment planning). The evidence is internally consistent though it is limited to synthetic data.
- The authors present segment‑wise performance curves for five canonical CSO policies. Different policies win in different segments and at different data regimes directly supports the intuition that “one size doesn’t fit all.”
- The paper argues that averaging policies can violate hard constraints, whereas selecting one feasible policy preserves feasibility. This is a good motivation for hard selection in constrained CSO and is consistent with the chosen benchmarks.

**Requested Changes:**

1. (Nice to have) Compare against a projected policy‑averaging baseline where averaged decisions are projected onto the feasible set (when projection is well‑defined), or explain carefully why such a comparison is invalid for the studied problems.
2. (Nice to have) Report compute metrics (training wall‑clock per N, number of optimization solves, memory, and inference latency for the meta‑policy and base policies) and discuss scalability as M, K, R grow.

---

### Decision · Action_Editor_5FMC · 2026-01-08

**Recommendation:** Accept as is

**Audience:**

Yes

**Audience Explanation:**

The CSO problem and proposing a method to find an optimal policy is relevant to the ML research community.

**Claims And Evidence:**

Yes

**Claims Explanation:**

The paper proposes a framework to select a context dependent policy for contextual stochastic optimization problem. At first it proposes a way to build a library of policies and then gives a procedure to select a good enough policies.